# DORIS study: domestic violence in orthopaedics, a prospective cohort study at a Swedish hospital on the annual prevalence of domestic violence in orthopaedic emergency care

Karin Svensson Malchau ,[1] Eva-Corina Caragounis,[2] Mikael Sundfeldt[1]

¹Department of Orthopaedics, Insitute of Clinical Sciences, University of Gothenburg, Gothenburg, Sweden
²Department of Surgery, Institute of Clinical Sciences, University of Gothenburg, Gothenburg, Sweden

**Correspondence to**
Dr Karin Svensson Malchau;
karin.am.svensson@vgregion.se

## ABSTRACT

**Background** Domestic violence (DV) is a major problem which despite many efforts persists globally. Victims of DV can present with various injuries, whereof musculoskeletal presentation is common.

**Objectives** The DORIS study (**D**omestic violence in **OR**thopaed**IcS**) aimed to establish the annual prevalence of DV at an orthopaedic emergency department (ED) in Sweden.

**Design** Female adult patients with orthopaedic injuries seeking treatment at a tertiary orthopaedic centre between September 2021 and 2022 were screened during their ED visit.

**Setting** This is a single-centre study at a tertiary hospital in Sweden.

**Participants** Adult female patients seeking care for acute orthopaedic injuries were eligible for the study. During the study period, 4192 female patients were provided with study forms and 1366 responded (32.5%).

**Primary and secondary outcome measures** The primary outcome measure was to establish the annual prevalence of injuries due to DV and second, to establish the rate of current experience of any type of DV.

**Results** One in 14 had experience of current DV (n=100, 7.5%) and 1 in 65 (n=21, 1.5%) had an injury due to DV.

**Conclusions** The prevalence of DV found in the current study is comparable to international findings and adds to the growing body of evidence that it needs to be considered in clinical practice. It is important to raise awareness of DV, and frame strategies, as healthcare staff have a unique position to identify and offer intervention to DV victims.

## STRENGTHS AND LIMITATIONS OF THIS STUDY

⇒ This is a prospective observational study investigating the annual prevalence of domestic violence (DV) in female orthopaedic patients using questionnaires containing validated questions for DV.

⇒ Study participants were approached individually without the presence of company and great discretion was taken to ascertain the safety of DV victims.

⇒ The study was designed to screen all female patients consecutively, and although difficulties in the practical implementation of the screening programme impeded the desired inclusion rate, a large volume of patients were included.

⇒ Study participants could not choose to be anonymous which may have deterred some patients from filling out the study questionnaires.

## INTRODUCTION

Domestic violence (DV) is a serious public health problem estimated to affect as many as 27% of women in partner relationships during their lifetime.[1] It is an insidious process, starting off with phases of systematic psychological abuse often leading to physical abuse.[2] Aside from its societal and individual economic consequences,[3] it is one of the most common causes for physical injuries

in women and victims are at great risk for mental health issues, suicide and homicide.[4] 20%–50% of female homicides are caused by a former or current intimate partner,[5 6] and in Sweden, the death toll due to known DV was 13 in 2020.[7]

Musculoskeletal injuries are one of the most common presentations of DV.[8 9] One in 50 women present to fracture clinics with an injury due to DV.[10] Recognition of DV as an injury mechanism is important and orthopaedic units have been suggested ideal for screening.[11 12] However, the difficulties of identifying DV are many. Victims may be prevented from seeking medical attention by their abuser which was found true for 36% of women in Canada.[8] A further challenge is the absence of active questioning in healthcare and that patients may not disclose occurrence of abuse.[13] Orthopaedic surgeons underestimate the prevalence of DV[14] and do not ask about DV.[10]

Implementation of screening within healthcare may lead to a greater detection of DV,

which in turn can be potentially lifesaving. Nevertheless, questioning for DV is not standard and formal documentation is poor.[15] Sweden is considered the most gender-equal country in the European Union[7]; however, research on DV in orthopaedics is scarce and little is known about its prevalence in Sweden. The current project aimed to identify the annual prevalence of orthopaedic injuries caused by DV and current experience of DV, in female patients at the largest orthopaedic emergency department (ED) in Sweden. Types of DV, injury due to DV and stated injury mechanisms were also evaluated.

## METHODS

### Study design

This is a self-reported questionnaire-based study including questions validated for detection of partner violence in an orthopaedic setting.[11]

### Objectives

The primary objective was to identify the annual prevalence of orthopaedic injuries sustained directly due to DV. The secondary objectives were to establish the annual prevalence of current experience of DV and investigate which types of DV, injuries and stated injury mechanisms were most common.

### Setting

The study was conducted at the ED of the Sahlgrenska University Hospital/Mölndal in Gothenburg, Sweden from 21 September 2021 to 21 September 2022. The ED averages 45 000 unique attendances yearly and the orthopaedic section has an average of 38 female attendances daily.

Sets of study information, marked with name and social security number, were assembled on triage. Staff were instructed to hand out the forms to all female patients fulfilling the study inclusion criteria. Forms were handed out in the examination room, filled out in private and put in a sealed envelope (figure 1). ED staff were unaware of status of study participation. The forms were contained inside the ED as a precautious measure to diminish the risk of unauthorised persons identifying potential victims. If ED staff discovered a case of DV when informing patients about the study, they were asked to mark the envelope with an 'X'. However, the patient was only included in the further analysis if she consented to study participation. Medical records of consenting patients reporting DV were reviewed to assess injury type and severity.

Patients who wished to meet a project counsellor were booked for a medical follow-up without mention of the counsellor. This was intentional to protect the patient in cases of cohabitation with the abuser.

### Participants

Patients of female sex of at least 18 years of age and with residency in Sweden triaged to the orthopaedic section of the ED were included in the study. Patients accompanied by someone, or with cognitive impairment or physical impairment, that is, dementia or poor eyesight, were excluded. Furthermore, patients who could not understand Swedish, English or Arabic were also excluded. No sample size calculation was conducted as the objective was to establish the annual prevalence of DV victims.

### Study questionnaire

Screening was performed using paper questionnaires, which had been developed based on the work of Sprague *et al*, where the direct questioning approach detected DV to a greater extent than other tools evaluated for orthopaedic use.[11] Additional questions on demography were

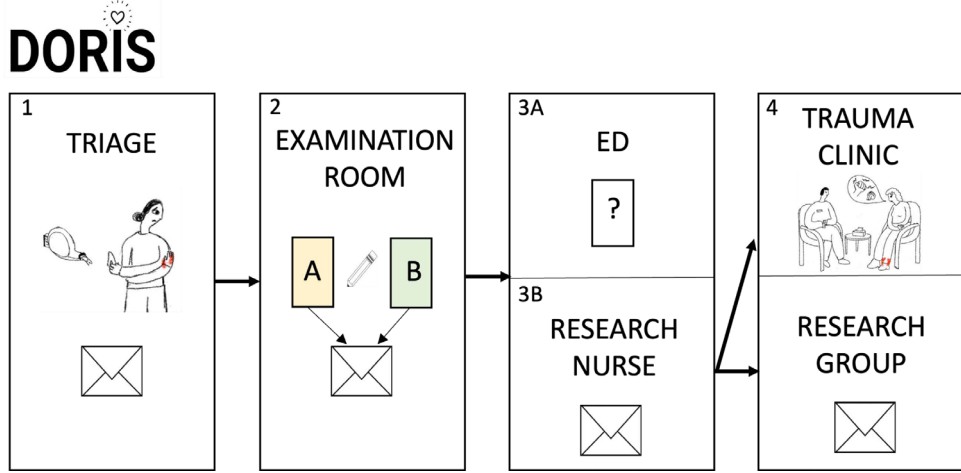

**Figure 1** The organisation of data collection. Study packages were assembled on triage (1) and patients were asked to fill out the forms A and B in private in the examination room and put them in a sealed envelope (2). Forms containing questions and study information were recollected and stored in the emergency department (ED) (3A) in order to diminish spread of word about the study. Sealed envelopes containing forms A and B were collected daily by the study research nurse (3B). The research nurse identified patients who wished to meet a welfare officer and booked them to the trauma clinic before data were inputted by the research group (4).

added (see online supplemental material). There were two forms (A and B) of which B was simplified and more anonymous in order to encourage higher responder rates (online supplemental material S1). Participants received both forms and could choose which form to fill in. Study forms were provided in Swedish and translated two-way in English and Arabic.

### Definitions

DV was defined as emotional, physical or sexual abuse. Any occurrence within the family, domestic unit or by former intimate partners, was included, as defined by the Istanbul Convention (2011).[16] A relationship was defined as a partnership lasting at least 1 month.

### Data analysis

Data were analysed descriptively with frequency counts and percentages for categorical variables. Software IBM SPSS V.29 was used for data analysis.

### Patient and public involvement

It was not deemed appropriate to involve patients or the public in the design, or conduct, or reporting, or dissemination plans of our research.

## RESULTS

In total, 4192 (30.4%) out of 13801 unique female attendances registered at the orthopaedic section of the ED were given study forms. Of these, 1366 (32.6%) agreed to inclusion (figure 2). The majority of responders spoke Swedish (99.4%), did not live in a socially disadvantaged area (80.4%) and were in a relationship (62.2%) (table 1).

### Experience of DV

Of the 1366 patients, 100 patients (7.5%) had current experience of DV and 21 (1.5%) of them had an injury due to DV. Of the 21 patients, 16 consented to filling out the study forms. The remaining five patients disclosed DV to healthcare staff but declined to fill out the study forms. Therefore, they were not included in the further analysis, leaving 95 patients of the 100 patients who had stated current experience of DV, eligible for further analysis (figure 2).

DV (any type) was reported by 89 (89/1361, 6.5%) patients in their current relationship. Emotional abuse was most common (69/89, 77.5%) followed by physical abuse (33/89, 37.1%) and sexual abuse (19/89, 21.3%) (figure 3).

### DV as a direct cause of injury

In total, 21 patients with an injury due to DV were identified (figure 2), meaning that 1 in 65 patients needed medical attention due to physical abuse. Of the 16 consenting DV victims, 8 had previously been in contact with healthcare for an injury due to abuse. Formal documentation of DV

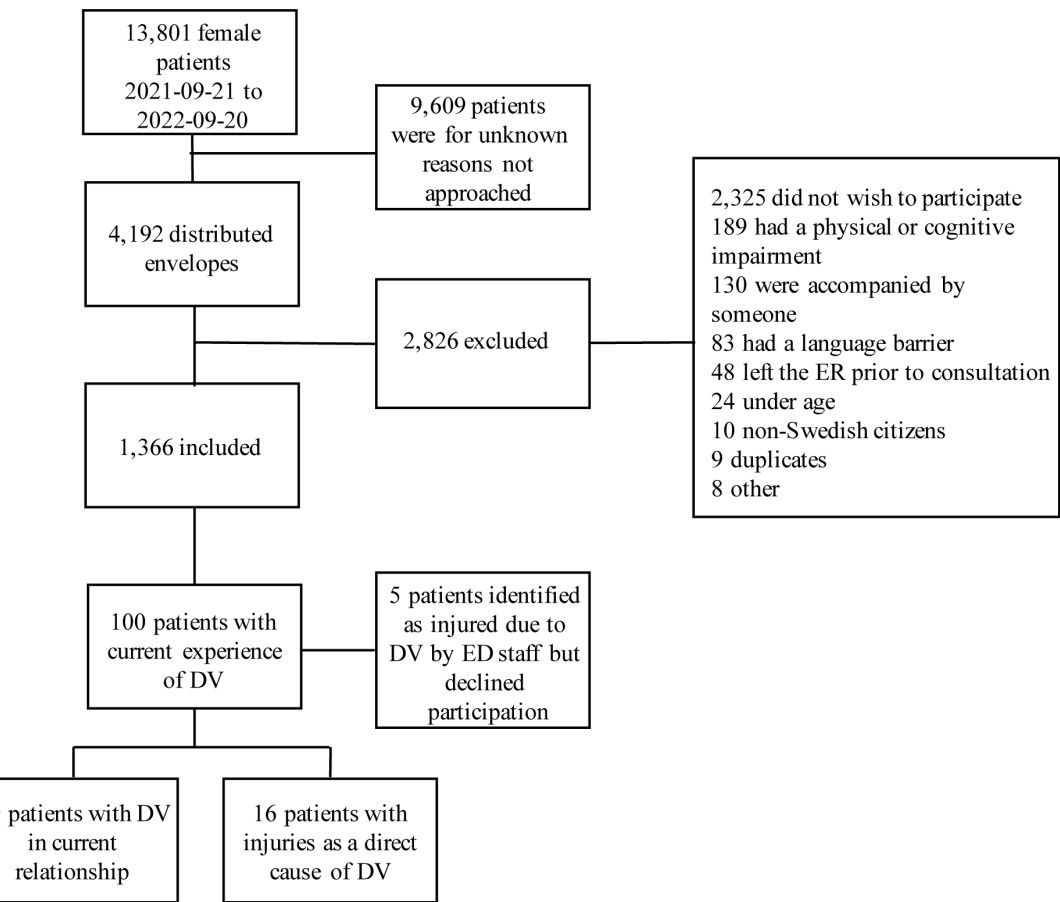

**Figure 2** Flow chart of study inclusion. DV, domestic violence; ED, emergency department.

**Table 1** Demographics of all responders and whether healthcare should ask about DV grouped by experience of DV

| | All responders (n=1361) | Responders reporting no experience of DV in current relationship (n=1165) | Responders reporting current DV, not DV as direct cause of injury (n=79) | Responders reporting DV as direct cause of injury (n=16) | Responders with missing or incomplete answers on current DV (n=101) |
|---|---|---|---|---|---|
| **Age (range, years)** | | | | | |
| 18–29 | 226 (16.6) | 201 (17.3) | 8 (10.1) | 2 (12.5) | 15 (14.8) |
| 30–39 | 211 (15.5) | 182 (15.6) | 13 (16.5) | 4 (25.0) | 12 (11.9) |
| 40–49 | 211 (15.5) | 174 (14.9) | 17 (21.5) | 4 (25.0) | 16 (15.8) |
| 50–59 | 262 (19.3) | 234 (20.1) | 15 (19.0) | 1 (6.3) | 12 (11.9) |
| 60–69 | 239 (17.6) | 199 (17.1) | 17 (21.5) | 4 (25.0) | 19 (18.8) |
| >70 | 202 (14.8) | 167 (14.3) | 8 (10.1) | 1 (6.3) | 26 (25.7) |
| Missing | 10 (0.7) | 8 (0.7) | 1 (1.3) | 0 (0.0) | 1 (1.0) |
| **Resident of a socially disadvantaged area** | | | | | |
| Yes | 227 (16.7) | 184 (15.8) | 16 (20.3) | 3 (18.8) | 24 (23.8) |
| No | 1101 (80.9) | 954 (81.9) | 58 (73.4) | 13 (81.3) | 75 (75.2) |
| Protected person/not a resident in Gothenburg | 9 (0.7) | 6 (0.1) | 3 (3.8) | 0 (0.0) | 0 (0.0) |
| Missing | 24 (1.8) | 21 (1.8) | 2 (2.5) | 0 (0.0) | 1 (1.0) |
| **Language** | | | | | |
| Swedish | 1353 (99.4) | 1161 (99.7) | 77 (97.5) | 15 (94.1) | 100 (99.0) |
| English | 5 (0.4) | 3 (0.3) | 1 (1.3) | 0 (0.0) | 1 (1.0) |
| Arabic | 3 (0.2) | 1 (0.0) | 1 (1.3) | 1 (5.9) | 0 (0.0) |
| **Education level** | | | | | |
| Compulsory school | 104 (7.6) | 82 (7.0) | 5 (6.3) | 2 (12.5) | 15 (14.9) |
| High school | 459 (33.7) | 395 (33.9) | 30 (38.0) | 9 (56.3) | 25 (24.8) |
| University | 727 (53.4) | 650 (55.8) | 43 (54.4) | 4 (25.0) | 29 (28.7) |
| Missing | 71 (5.3) | 38 (3.3) | 1 (1.3) | 1 (6.3) | 31 (30.7) |
| **Partner sex** | | | | | |
| No partner | 430 (31.6) | 427 (36.7) | 0 (0.0) | 3 (17.6) | 0 (0.0) |
| Male | 806 (59.2) | 711 (61.0) | 75 (95.0) | 10 (64.7) | 10 (10.0) |
| Female | 40 (2.9) | 22 (2.0) | 2 (2.5) | 2 (11.8) | 14 (13.9) |
| Missing | 85 (6.2) | 5 (0.4) | 2 (2.5) | 1 (5.9) | 77 (76.2) |
| **Duration of relationship** | | | | | |
| Less than 1 year | 36 (2.6) | 30 (2.6) | 4 (5.1) | 2 (12.5) | 1 (1.0) |
| 1–5 years | 165 (12.1) | 144 (12.4) | 12 (15.2) | 3 (18.8) | 6 (5.9) |
| 6–10 years | 96 (7.1) | 80 (6.9) | 12 (15.2) | 2 (12.5) | 2 (2.0) |
| More than 10 years | 567 (41.7) | 476 (40.9) | 50 (63.3) | 5 (31.3) | 36 (35.6) |
| No partner | 440 (32.3) | 430 (36.9) | 1 (1.3) | 3 (18.8) | 6 (5.9 |
| Missing | 57 (4.2) | 5 (0.4) | 0 (0.0) | 2 (12.5) | 50 (50.0) |
| **Have you ever sought medical care for DV?** | | | | | |
| No | 1050 (77.1) | 942 (80.9) | 57 (72.2) | 8 (50.0) | 43 (42.6) |
| Yes | 54 (4.0) | 38 (3.3) | 10 (12.7) | 3 (18.8) | 4 (4.0) |
| Missing | 257 (18.9) | 185 (15.9) | 13 (16.5) | 5 (31.3) | 54 (53.5) |
| **Should healthcare workers ask about DV?** | | | | | |
| Yes | 1209 (88.8) | 1068 (91.7) | 76 (96.2) | 14 (87.5) | 51 (50.5) |
| No | 41 (3.0) | 30 (2.6) | 2 (2.5) | 0 (0.0) | 9 (8.9) |
| Missing | 111 (8.2) | 67 (5.8) | 1 (1.3) | 2 (12.5) | 41 (40.6) |

DV, domestic violence.

was noted in eight medical records, and in the remaining cases the injury mechanism was unspecified fall trauma (table 2).

The age span of DV victims was 18–76 years. Three patients were from socially disadvantaged areas and three patients had female partners. The majority of patients

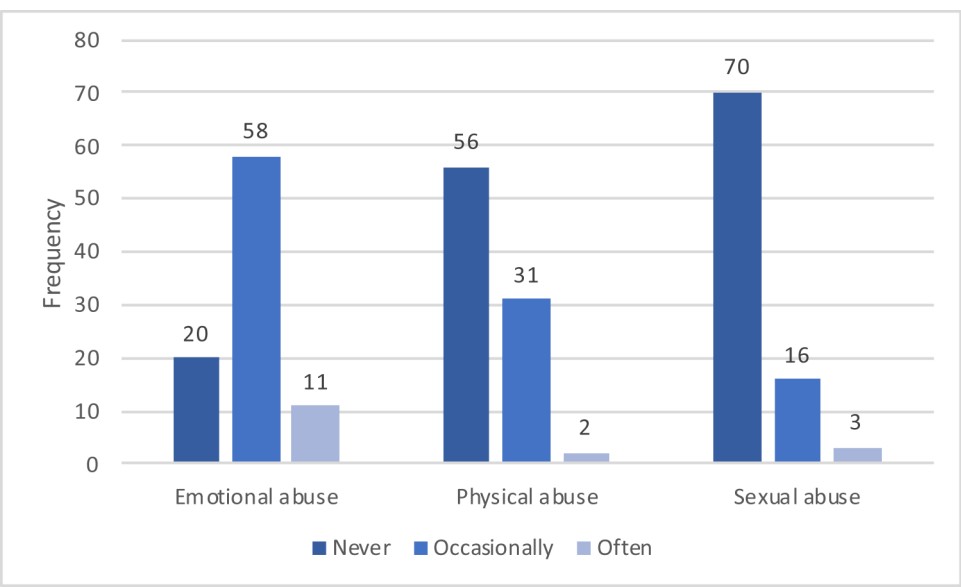

**Figure 3** Occurrence and type of abuse among patients reporting on DV in a current relationship. Note: 3 of the 95 patients reporting on DV were not in a current relationship and an additional 3 patients did not fill in the questions about abuse in their relationship.

had completed high school but had no further academic education (table 1). Eight patients reported on repeated abuse in their current relationship of which five stated an occurrence of both emotional, physical and sexual abuse.

Fractures were the most prevalent injury followed by contusions and joint distortions (table 3). Five patients sustained injuries requiring sick leave and two patients required surgery (table 3). Thirty-seven follow-up visits were recorded due to DV injuries (excluding visits to the counsellor).

### Screening for DV

In total, 1208 women (89.0%) were of the opinion that healthcare staff should ask about DV (table 1). However, 2 of the 16 patients (12.5%) injured due to DV did not feel that screening was necessary. Fifty-four patients (4.0%) had previously contacted healthcare for physical

**Table 2** Injury mechanism as stated in the medical records and treatment needs due to DV

|  | Frequency n (%) |
| --- | --- |
| Formal documentation of DV in medical record |  |
| Yes | 8 (50) |
| No | 8 (50 |
| Stated injury mechanism in medical record |  |
| Fall trauma, unspecified | 8 (50) |
| Abuse | 8 (50) |
| Orthopaedic treatment |  |
| Pain medication and physiotherapy | 8 (50) |
| Immobilisation (cast/orthosis) | 6 (38) |
| Surgery | 2 (12) |
| Need for sick leave |  |
| Yes | 5 (31) |
| No | 11 (69) |
| DV, domestic violence. |  |

**Table 3** Type of injuries noted in cases with DV as direct cause of injury

| Injury type and localisation | Frequency n (%) |
| --- | --- |
| Fracture | 6 (38) |
| Hand | 5 |
| Foot | 1 |
| Contusion | 4 (25) |
| Upper extremity | 1 |
| Lower extremity | 3 |
| Distortion | 4 (25) |
| Shoulder | 1 |
| Knee | 1 |
| Foot | 2 |
| Joint dislocation | 1 (6) |
| Ligament rupture | 1 (6) |
| Laterality of injury |  |
| Right | 10 (63) |
| Left | 4 (25) |
| Missing | 2 (12) |
| DV, domestic violence. |  |

abuse (table 1), whereof 34 of these patients were still in an abusive relationship.

The project counsellors had contact with 23 patients whereof 8 had been injured due to DV. 12 patients (52.2%) showed up for their appointment. Six patients failed to appear, four had misunderstood or were too injured to come for their appointment. One patient had given a faulty address and did not respond to phone calls.

## DISCUSSION

The DORIS study aimed to establish the prevalence of injuries directly caused by DV, current experience of DV, types of DV, injuries and stated injury mechanisms in female patients in the largest orthopaedic ED in Northern Europe. It also evaluated the rate of types of DV, injuries due to DV and what injury mechanisms were stated by victims. A rate of 1 in 14 patients (100/1366, 7.5%) with current experience of DV and 1 in 65 patients (21/1366, 1.6%) injured due to DV was established.

The prevalence of injuries due to DV (1.6%) is within the span of prevalence reported by the PRAISE (Prevalence of abuse and initimate partner violence surgical evaluation) group (0%–3%), which conducted a multi-national investigation of intimate partner violence in female patients at orthopaedic injury clinics.[10] Current experience of DV was recorded in the DORIS study whereas previous studies have investigated the 12-month prevalence. A 12-month prevalence of DV of 15%–22% in orthopaedic patients has previously been reported.[10 17] In the DORIS study, 6.5% (89/1366) experienced DV in a current partner relationship. Differences in recruitment methods, study settings and staff engagement could serve as explanations to the lower prevalence in Sweden. The lower prevalence may also reflect governmental and societal policies on gender equality in Sweden.

When comparing proportions of type of abuse, the present study established that emotional abuse was the most common. This is also true in Scottish, American and Canadian settings.[10 17] However, surprisingly, in the Netherlands and in Denmark, countries seemingly more comparable to Sweden, physical abuse was most common.[10] It may be difficult to understand what is meant by emotional abuse, the DORIS study forms contained examples of emotional abuse which may explain the higher prevalence.

Formal documentation of DV was noted in 50% of cases, meaning that 50% were not identified in the regular healthcare setting. Routine screening of DV leads to higher detection rates[18]; however, only 2% of healthcare workers in orthopaedics routinely ask about it.[19] Surgeons feel uncomfortable and unsure of what to do if their patient is a victim which calls for better education and support models within healthcare.[20 21]

Although it is important to be suspicious of inconsistent injury mechanisms or 'red flags', such as falling down the stairs,[22] feasible injury mechanisms were disclosed in 50% of the DV cases. Hence, questioning for DV should not just be conducted when suspicion is raised, as is often the case. Within the DORIS study, direct questioning, in questionnaire format, was used as this has proven efficient for DV screening and is less time consuming in an ED setting.[11 23] However, the study forms contained a lot of text due to regulations stated by the Ethical Review Board, which may have discouraged potential responders. In the continued work of improving DV detection at the study site, efforts will be made to optimise the screening tool.

Merely 50% of patients with an injury due to DV had previously been in contact with healthcare for DV. Hence, the remaining patients may have presented with an index injury. This finding supports the, previously suggested,[11 19] need for screening in orthopaedic settings, as early intervention can be potentially lifesaving. Up to 81% of female patients are of the opinion that healthcare staff should ask about DV.[10 19 23] The corresponding numbers were somewhat higher in the DORIS study (89% in the entire cohort and 94%–96% in abused patients). Cultural differences and thereby expectations on healthcare may explain the aforementioned variances.

The strength of the DORIS study is its setting at the largest orthopaedic ED in Northern Europe. After, the PRAISE study,[10] DORIS is the largest prevalence study in orthopaedics. Due to COVID-19 restrictions during the study period, company was generally not allowed in the ED which facilitated the distribution of study forms. Victims of DV were also offered follow-up with a counsellor within the study.

A major limitation may be non-response bias. Although the study was regarded as important by ED staff, the distribution rate of study forms was 30% and response rate 33%. The authors had meetings with ED staff and two counsellors were recruited to provide an in-house support programme to increase the likelihood for staff engagement.[19] Unfortunately, due to management issues, the staffing situation became more turbulent with several experienced nurses and assistant nurses choosing to resign throughout the year. The authors believe that the inconsistencies in staffing were the main reason for poor study enrolment (online supplemental material 1). In addition, despite being an excellent forum for DV screening,[18] in regard to the 'open window phase' (in which victims may be more receptive and prone to seek help after abuse),[24] the ED as such is a busy and stressful place. In general, detecting DV may be difficult in such a setting: staff may be unaware of DV as a problem, and patients may feel uncomfortable confiding in ED staff. For this reason, it is crucial to structure EDs in a manner where triage can be done in private, as also suggested by Ahmad et al,[18] and where patients are unaccompanied in triage as standard routine.

Poor response rate was partly expected. Similar studies[10 17] have had different approaches to recruitment making it difficult to evaluate what an acceptable response rate is. Due to the delicate nature of the study, the authors had preferred that social security number and further personal details were omitted when

consenting to the study. The need to do this may have deterred potential victims from disclosing DV. However, full disclosure of personal details was a requirement from the Ethical Review Board due to research regulations. Furthermore, the authors have reason to believe that the 2325 patients who for some reason did not wish to participate in the study may not have received proper study information or been given a chance to fill out the study forms.

The exclusion criteria imply certain limitations. Elderly patients, either accompanied by caregivers or with the diagnosis of dementia, were not included. Despite the difficulties of capturing cases in this group, it is important to acknowledge their vulnerability and that both dementia and female sex are predictive of abuse.[25] Furthermore, the authors acknowledge that DV affects both female and male patients. Screening of females was chosen as female DV patients have a greater fracture risk, 83% of ED visits due to DV are female and 50% of female homicides are due to DV.[12] However, the long-term goal for the DORIS project is to provide a healthcare programme dedicated to DV patients regardless of sex.

The DORIS study focused on current abuse, whereas previous research, such as conducted by the PRAISE group and Sardinha *et al*, also investigated life-time abuse.[1 10 17] In hindsight, the inclusion of life-time abuse would have been interesting for comparative reasons. However, when designing the study, the authors decided that the patient's current situation was the most clinically relevant and therefore most important.

Despite its limitations, and a probable under-reporting of DV, the finding of 1 in 65 patients translates to 1 victim of DV injuries nearly every second day, and 2–3 patients with current experience of DV daily, at the study centre. Interventions are essential to disrupt continued abuse and healthcare has an important role in the detection of DV.[9 26] The experience generated by the present study suggests that screening is necessary in order to improve identification of DV cases and that patients expect healthcare to engage in detecting DV. The results from the DORIS study will be used to improve routines at the study site, and hopefully inspire to similar actions elsewhere.

## CONCLUSION

The prevalence of DV established in the current study implies a high annual volume of DV victims at the study site. DV victims may come to an orthopaedic setting with an index injury and healthcare staff have a unique opportunity to intervene. The DORIS study adds to the growing body of evidence that DV needs attention in the healthcare setting. Increased awareness and actions to identify DV are imperative, and it is important to educate, engage and provide adequate conditions for healthcare staff to conduct screening. Future work should focus on implementing DV screening as a routine and provide a safe environment for DV victims in all healthcare disciplines.

**Acknowledgements** The authors would like to thank Sandra Rosnell, Linda Stolpe, Stella Sundfeldt, Ann-Christin von Corswant, Anne Louise Gidestrand, the medical staff at the ED at Mölndal's Hospital and all the patients who chose to participate.

**Contributors** KSM participated in the planning and design of the study, collection of patient data, analysed the data, interpreted the data, drafted the manuscript and critically revised the manuscript. E-CC and MS participated in the planning and design of the study and critically revised the manuscript. KSM is the guarantor of this study.

**Funding** The study was funded by Doktor Felix Neuberghs stiftelse N/A, BGS forskningsstipendium N/A, Göteborgs Läkaresällskap N/A, SU-fonderna N/A and Konrad och Helfrid Johanssons stiftelse N/A.

**Competing interests** None declared.

**Patient and public involvement** Patients and/or the public were not involved in the design, or conduct, or reporting, or dissemination plans of this research.

**Patient consent for publication** Not applicable.

**Ethics approval** This study involves human participants and was approved by The Swedish Ethical Review Board (DNR 2021-01752). Participants gave informed consent to participate in the study before taking part.

**Provenance and peer review** Not commissioned; externally peer reviewed.

**Data availability statement** No data are available. All data relevant to the study are included in the article or uploaded as supplementary information.

**ORCID iD**
Karin Svensson Malchau http://orcid.org/0000-0001-9050-7929

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
