## [Reviewer comments · BMJ Open]

ARTICLE DETAILS

TITLE (PROVISIONAL)	The DORIS study: Domestic violence in ORthopaedicS, a prospective cohort study at a Swedish hospital on the annual prevalence of domestic violence in orthopaedic emergency care
AUTHORS	Svensson Malchau, Karin; Caragounis, Eva-Corina; Sundfeldt, Mikael

VERSION 1 – REVIEW

REVIEWER	Sheila Sprague McMaster University
REVIEW RETURNED	06-Mar-2024

GENERAL COMMENTS	This is a well written and well conducted cross-sectional study. The largest limitation is the enrollment rate, which may lead to an over or under estimate of the prevalence of IPV. Please consider using the language of intimate partner violence instead of domestic violence throughout the manuscript. In the objectives, consider revising year prevalence to annual prevalence. Were participants who spoke English and/or Arabic included (this wording is unclear)? This should be written as an inclusion/exclusion criteria instead of a sentence about the questionnaire translation. It would be helpful to have a section on the development of the questionnaire. The aims at the end of the introduction, the objectives in the middle of the methods, and the results do not align. The methods section could be better organized for flow and content. For example, the questionnaire administration is described under the setting; the translation of the questionnaire is under the participants section; the participants section should go ahead of the description of the completion of the questionnaires; the objectives is in the middle of the methods section (would go better at the beginning). The results section is also difficult to follow in places. Please state annual prevalence. It cannot determine where the number 95 came from in line 177. The results include data that are not listed as objectives. This should align.
---

	The second paragraph of the discussion is unclear. Please rework.
--	---

REVIEWER	Jennifer A. Kunes Columbia University Irving Medical Center, Orthopaedic Surgery
REVIEW RETURNED	12-Mar-2024

GENERAL COMMENTS	Overall, this article is a great addition to the current literature around DV/IPV in the orthopaedic patient population. In this large prospective study, 4,192 patients were invited to complete DV screening forms and 1,366 patients responded. The reported prevalences of DV and DV-related injuries are in general agreement with prior published studies, e.g. PRAISE 2013. This article is acceptable for publication with the minor revisions detailed below. Abstract: Overall representative of the study. Line 52: Please clarify this sentence - were the prior documented cases (before study involvement) identified due to prior screening, or does this refer to the present study screening? Introduction: Line 88: Is there existing data to express the death toll of DV relative to population? (e.g. annual cases per X number of people)? Line 94: Please clarify factors preventing survivors of DV from seeking medical attention - does this refer to prevention by the abusive partner? Methods: Line 124: How were the two forms of the survey distributed? Did all patients physically receive both forms, or were participants randomly assigned one of the two forms? Results: Line 174: Please explain the rationale for including the 5 patients who declined study participation and reported DV directly to providers in the calculation of prevalence of DV. Are these 5 patients accounted for in the denominator of 1,366 included patients, as they declined to be included? Line 175: What percent of patients responded to each of the two forms? What was the prevalence of DV detected through one versus the other - was there a difference? Line 203: Can the authors comment on the demographic characteristics of those that did versus did not believe that healthcare staff should ask about DV? Discussion: Line 247: As above, can the authors comment on the relative response rates and detection rates between the two forms of the questionnaire? Line 257: As above, is there data to suggest a demographic difference between those who do believe healthcare staff should screen versus those who do not? Conclusion: Well-written. Supplementary materials:
---

	How were the 101 respondents with missing or incomplete answers on current DV exposure accounted for in the analysis?
--	---

VERSION 1 – AUTHOR RESPONSE

The largest limitation is the enrollment rate, which may lead to an over or under estimate of the prevalence of IPV.

We very much agree that this is a great limitation.

Please consider using the language of intimate partner violence instead of domestic violence throughout the manuscript.

Thank you for your comment. We have considered this throughout the process of both planning and conducting the study. However, we chose to adopt the definition of domestic violence according to the Istanbul convention as it also includes violence within the family and domestic unit. We did not want to exclude patients who were victims to violence by a non-intimate close person. Please see the definition in the methods section.

In the objectives, consider revising year prevalence to annual prevalence.

Thank you for your input on this. This has now been revised in the abstract, introduction and methods section.

Were participants who spoke English and/or Arabic included (this wording is unclear)? This should be written as an inclusion/exclusion criteria instead of a sentence about the questionnaire translation.

We apologize for any confusion and have revised this. Hopefully it will be more clear now. Patients who spoke English or Arabic were included in the study. Please see lines 139-140 in the section about participants in the methods section.

It would be helpful to have a section on the development of the questionnaire.

This has now been added. Please see lines 144-150 in the section about participants in the methods section.

The aims at the end of the introduction, the objectives in the middle of the methods, and the results do not align.

Thank you for observing this. This has now been altered.

The methods section could be better organized for flow and content. For example, the questionnaire administration is described under the setting; the translation of the questionnaire is under the

participants section; the participants section should go ahead of the description of the completion of the questionnaires; the objectives is in the middle of the methods section (would go better at the beginning).

Thank you for this input. We have re-structured the methods section and hope that it will be easier to follow.

The results section is also difficult to follow in places. Please state annual prevalence. It cannot determine where the number 95 came from in line 177.

We hope that it is easier to understand the number 95 and to follow the results section after our revision.

The results include data that are not listed as objectives. This should align.

Please see our revised objectives.

The second paragraph of the discussion is unclear. Please rework.

We have edited the paragraph and hopefully clarified the message.

Reviewer: 2

Dr. Jennifer A. Kunes, Columbia University Irving Medical Center

Comments to the Author:

Overall, this article is a great addition to the current literature around DV/IPV in the orthopaedic patient population. In this large prospective study, 4,192 patients were invited to complete DV screening forms and 1,366 patients responded. The reported prevalences of DV and DV-related injuries are in general agreement with prior published studies, e.g. PRAISE 2013. This article is acceptable for publication with the minor revisions detailed below.

Abstract: Overall representative of the study.

Line 52: Please clarify this sentence - were the prior documented cases (before study involvement) identified due to prior screening, or does this refer to the present study screening?

We apologize for the confusion, the documented cases were identified thanks to the present study screening. Due to a word count limitation in the abstract after revision requested by the journal, this sentence has been omitted.

Introduction:

Line 88: Is there existing data to express the death toll of DV relative to population? (e.g. annual cases per X number of people)?

To our knowledge there is no Swedish data on death toll due to DV relative to population. Its statistic is expressed as an annual number.

Line 94: Please clarify factors preventing survivors of DV from seeking medical attention - does this refer to prevention by the abusive partner?

Thank you for helping us clarify this. It refers to the prevention by the abusive partner. This has now been added. Please see line 96.

Methods:

Line 124: How were the two forms of the survey distributed? Did all patients physically receive both forms, or were participants randomly assigned one of the two forms?

This has now been further explained, please see line 149.

Results:

Line 174: Please explain the rationale for including the 5 patients who declined study participation and reported DV directly to providers in the calculation of prevalence of DV. Are these 5 patients accounted for in the denominator of 1,366 included patients, as they declined to be included?

Yes, they are included in the denominator. They did not fill out demographic data in the questionnaire, and therefore we could not include them in our descriptive analysis. The demographic analysis is conducted on 1,361 patients.

Line 175: What percent of patients responded to each of the two forms? What was the prevalence of DV detected through one versus the other - was there a difference?

We had expected more patients who experienced DV to fill out the shorter form (B). Surprisingly, there was no greater difference in the response rate between the forms. Only two of the patients who had an injury due to DV chose to answer form B, and remaining patients filled out both forms. Patients who did not have an injury due to DV filled out both forms.

Line 203: Can the authors comment on the demographic characteristics of those that did versus did not believe that healthcare staff should ask about DV?

We have re-run analyses on these patients to see whether there were any differences between the two groups. Patients who did not believe it was within the scope of healthcare to screen for DV had a lower degree of education (where 22% had disclosed a university degree compared to 58% in the group of patients who felt it was important healthcare screen for DV). All patients (100%) who believed healthcare should not ask about DV understood and spoke Swedish. Other than that, the distribution of age and socioeconomic area of habitancy were similar.

Discussion:

Line 247: As above, can the authors comment on the relative response rates and detection rates between the two forms of the questionnaire?

Thank you for your question. The aim of the study was not to evaluate the detection rates between the two forms. However, two patients with injuries due to DV had chosen to just fill out the shorter questionnaire (B), the remaining patients had filled out both.

Line 257: As above, is there data to suggest a demographic difference between those who do believe healthcare staff should screen versus those who do not?

Please see our response above.

Conclusion: Well-written.

Supplementary materials:

How were the 101 respondents with missing or incomplete answers on current DV exposure accounted for in the analysis?

These patients were described in accordance with Table 1. Filling out the other questions of the forms, but not filling out questions about DV, is suspicious. However, we did not want to risk overestimating the prevalence of current DV, therefore, we chose to not make an assumption that these patients may have been exposed to DV. Instead they were not considered victims of DV.

VERSION 2 – REVIEW

REVIEWER	Sheila Sprague McMaster University
REVIEW RETURNED	10-Apr-2024

GENERAL COMMENTS	Thank you for carefully revising your manuscript based on reviewer comments. I do not have any further comments at this time.
---

REVIEWER	Jennifer A. Kunes Columbia University Irving Medical Center, Orthopaedic Surgery
REVIEW RETURNED	17-Apr-2024

GENERAL COMMENTS	The DORIS study is a prospective observational study that quantifies the prevalence of domestic violence among female patients in a Swedish emergency department. Prior studies have demonstrated large differences in detected prevalence of IPV/DV between retrospective and prospective studies, and prospective screening studies are more likely to represent the true prevalence. The findings of the DORIS study are in line with the PRAISE 2013 study. This study is suitable for publication and should be of interest to readers.
--